# Moderate Nitrogen Reduction Increases Nitrogen Use Efficiency and Positively Affects Microbial Communities in Agricultural Soils

**Jianghua Tang, Lili Su, Yanfei Fang, Chen Wang, Linyi Meng, Jiayong Wang, Junyao Zhang and Wenxiu Xu ***

College of Agriculture, Xinjiang Agricultural University, Urumqi 830052, China
* Correspondence: xjxwx@sina.com

**Abstract:** Excessive nitrogen fertilizer usage in agricultural often leads to negative ecological and production gains. Alterations in the physical and chemical properties and microbial community structure of agricultural soils are both the cause and consequence of this process. This study explored the perturbation of soil properties and microorganisms in agricultural soils by different nitrogen levels. Soil total nitrogen, total phosphorus, and total potassium decreased in the shallow soil layer with decreasing nitrogen. Changes in nitrogen affected soil organic matter, pH, bulk density, and water content. However, a moderate reduction in nitrogen did not cause significant yield loss; the increased nitrogen use efficiency was the main reason, attributed to the available phosphorus and potassium. Short-term changes in nitrogen had limited effects on soil microbial community structure. Bacteria were more susceptible to perturbation by nitrogen changes. Nitrogen reduction increased the relative abundance of MND1 (1.21%), RB41 (1.96%), and Sphingomonas (0.72%) and decreased Dongia (0.3%), Chaetomium (0.41%), and Penicillium (0.5%). Nitrogen reduction significantly increased the bacteria functional composition of aerobic ammonia oxidation (4.20%) and nitrification (4.10%) and reduced chemoheterotrophy (2.70%) and fermentation (4.08%). Available phosphorus specifically drove bacterial community structure variation in the shallow soil layers of moderate nitrogen reduction treatments. Steroidobacter, RB41, Gemmatimonas, Ellin6067, Haliangium, and Sphingomonas were the main component nodes in this community structure. These results provide insights into the study of nitrogen and microorganisms in agricultural soils.

**Keywords:** microbial diversity; soil nutrient; nitrogen cycle; co-occurrence network; bioinformatics

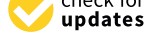



## 1. Introduction

Modern agriculture heavily relies on agricultural inputs rather than improvements in cropping efficiency. However, over-fertilization has resulted in degradation, erosion and structural damage of agricultural soils, posing a serious threat to global food security and climate stability. With the advancement of ecology and biotechnology, soil science research has transitioned from traditioned physico-chemical methods to multidimensional high-resolution histological analysis. This approach emphasizes the role of soil microorganisms in soil carbon and nitrogen cycling, nutrient turnover, and biosecurity [1–3]. Next-generation sequencing technology, based on amplicon sequencing of genetically conserved regions of microorganisms, provided a global view of the overall changes in microbial community structure in soils, enabling the identification of microbial functions at the genetic level.

Maintaining a stable soil microbial community structure is crucial in preserving soil production potential and overall soil health. Various nitrogen (N) addition strategies induced changes in soil physicochemical properties which are correlated with alterations in microbial community structure and partial microbial abundance, thereby displaying differences in carbon and N use efficiencies [4]. The formation of a suitable microbial community structure can enhance N fertilizer utilization and result in increased yields

while reducing N fertilization [5]. However, long-term excessive and irrational application of nitrogen fertilizers can deteriorate the microbial community structure in agricultural soils, hindering the development of soil ecosystems [6–8]. Various microorganisms in soil interact with each other and environmental factors, forming complex microbial networks that promote or inhibit the uptake of soil nutrients by crops through multiple biochemical and metabolic pathways [9,10]. Therefore, investigating the response mechanism of soil microorganisms to N fertilizer application is crucial in guiding the application of N fertilizer. This study comprehensively identifies the changes in bacterial and fungal species and abundance in soil layers at different depths under gradient nitrogen reduction using 16 s and ITS high-throughput sequencing. The Mantel test, redundancy analysis, and Spearman correlation analysis were used to explore the response mechanism of different soil layers to nitrogen fertilizer application reduction and reveal the core genera in the variation of soil microbial community structures. The aim is to provide a scientific and rational response to nitrogen fertilizer application in agricultural production, which will offer theoretical support for scientific and rational reduction of N fertilizer application.

## 2. Materials and Methods

### 2.1. Study Area

The experiment was conducted at the cotton breeding base of Xinjiang Agricultural University ($43°20'$–$45°20'$ N, $84°45'$–$86°40'$ E) in 2020 and 2021. The region has a typical temperate continental climate with an annual rainfall ranging from 125–207.7, and annual evaporation ranging from 1000–1500 mm. Agricultural production in the area depends on surface water irrigation. The average annual temperature is 7.4–8.1 °C, with annual sunshine hours of 2318–2732 h and a frost-free season of 146–189 d. The soil of the test farmland is loamy from 0–60 cm and sandy from 60–80 cm. The sample plots were cultivated using conventional cotton (*Gossypium hirsutum* L.) cultivation practices, and no additional treatments were applied. Regular manual weeding was performed. The soil's base nutrients were determined before the experiment in 2020, specifically from a depth of 0–40 cm.

### 2.2. Experimental Treatment and Soil Sampling

Referring to the published research, this experiment involved four different levels of reducing nitrogen fertilization (RNE) treatment. The RNE treatments included nitrogen application (375 kg/hm$^2$, CK), 20% conventional nitrogen reduction (300 kg/hm$^2$, N2), 40% conventional nitrogen reduction (225 kg/hm$^2$, N4) and 100% conventional nitrogen reduction (0 kg/hm$^2$, N0). Urea containing 46% N was used as the N fertilizer material, and all fertilizer applications were made at different stages of cotton growth. These were 5% at the bud stage, 25% at the pre-bell stage, 35% at the mid-bell stage, 30% at the post-bell stage, and 5% at flocculation stage. Calcium superphosphate (450 kg/hm$^2$, 46% P$_2$O$_5$) was applied before sowing, while potassium phosphate (150 kg/hm$^2$, 52% P$_2$O$_5$, 34% K$_2$O) was applied during cotton reproduction. Each treatment was replicated three times.

Soil samples were collected after the cotton boll completely opens, from between the rows of narrow rows in each plot. Dead branches and weeds were removed from the soil surface before sampling. Soil profiles were dug with a 5 cm diameter soil corer, and soil was taken in layers from 0–20 cm and 20–40 cm. The soil was then flattened with a sharp knife. The physicochemical traits of the soil were determined, and next-generation sequencing was performed. The soil samples were immediately transported to the laboratory for processing in ice boxes.

### 2.3. Soil Properties and Nutrient Determination

Soil bulk density was determined using a cutting ring method. Soil water content was calculated by measuring the difference between dry weight and fresh weight. Yield and yield components were measured by manual harvesting. Total nitrogen (TN) was assayed using the indophenol blue colorimetric method. Total phosphorus (TP) was

assayed using the HClO$_4$-H$_2$SO$_4$ method. Total potassium (TK) was assayed using the flame photometric method. NH$_4^+$-N was extracted using 2 mol/L KCl and assayed using the indophenol blue colorimetric method. Nitrate nitrogen (NO$_3^-$-N) was measured using the dual-wavelength UV spectrophotometry method. The available phosphorus (AP) was extracted using 0.5 mol/L NaHCO$_3$ and assayed using the molybdenum blue colorimetric method. Available potassium (AK) was extracted using NH$_4$OAc and measured using flame photometry. pH was determined using potentiometric titration. Soil organic matter was assayed using the potassium dichromate-external heating method.

### 2.4. 16 S rRNA and ITS Gene Sequencing

In this experiment, soil genomic DNA was extracted using the MN NucleoSpin 96 Soil kit (Macherey-Nagel, Duren, Germany). The bacterial 16s rRNA (V3 + V4) and fungal ITS1 conserved region were amplified using specific primers. The bacterial amplification primers used were 338F (5'-ACTCCTACGGGAGGCAGCA-3') and 806R (5'-GGACTACHVGGGTWTCTAAT-3'). The following fungal amplification primers were used: ITS1F (5'-CTTGGTCATTTAGAGGAAGTAA-3') and ITS2 (5'-GCTGCGTTCTTCATCGATGC-3'). The reaction system was 10 μL, and the reaction conditions were as follows: pre-denaturation 95 °C for 5 min, denaturation at 95 °C for 30 s, annealing at 50 °C for 30 s, extension at 72 °C for 40 sec for 20 cycles, followed by extension at 72 °C for 7 min to terminate the reaction and storage at 4 °C. The PCR products were quantified by electrophoresis, mixed in equal mass ratio, and recovered by cutting 1.8% agarose gel. High-throughput sequencing was performed using the Illumina Hiseq platform. Additionally, the accession number of all raw data in National Center for Biotechnology Information was PRJNA860158.

### 2.5. Data Analysis

OTU clustering was performed using USEARCH [11] on sequences with more than 97% similarity. Species annotation was carried out using the RDP Classifier [12], with the Silva database used for bacteria and the Unite database for fungi. Alpha diversity indices were calculated using Mothur software v.1.48.0. [13]. The functional composition of bacteria and fungi was determined using FAPROTAX [14] and FUNGuild [15], respectively. Beta diversity analysis, multiple correlation analysis, and grey relational analysis were conducted using R scripts. Redundancy analysis was performed using the R package vegan, and co-occurrence network graphs were generated using Gephi v.0.10.1 [16]. Statistical analysis and significance tests were carried out using Prism v.9.5.1. and SPSS v.27.

## 3. Result

### 3.1. Soil Properties and Correlation Analysis

The application RNE resulted in changes in various soil physical and chemical properties (Figure 1A). Soil organic matter, bulk weight, soil NH$_4^+$-N, and TK showed a decreasing trend with N fertilizer reduction in both the surface layer (0–20 cm) and the deep layer (20–40 cm), followed by an increasing trend. Conversely, soil pH, water content, TP, AP, and AK exhibited an increasing trend first and then a decreasing trend in each soil layer. The levels of nitrate and ammonium nitrogen in all soil layers decreased with decreasing N fertilizer use. Compared to conventional N (CK), the 20% N reduction treatment reduced soil bulk density, TN, and TK content, while it increased soil TP, AP, and AK content. In the 0–20 cm soil layer, these three nutrients were 4.04%, 44.58%, and 9.94% higher than the CK, respectively, and in the 20–40 cm soil layer, they were 2.05%, 44.47%, and 22.60% higher than the CK.

The correlations between different soil nutrients varied layers among the different soil layers (Figure 1B). In the 0–20 cm soil layer, N reduction showed a positively correlated with TP, TK, NO$_3^-$-N, NH$_4^+$-N, and AK, with R$^2$ values ranging from 0.737 to 0.996. In contrast, AP showed a significantly correlated with RNE. In the 20–40 cm soil layer, further analysis of the relationship between N reduction and various indicators indicated a positive correlation with all indicators except TN. The correlation coefficients ranged from

0.116–0.976. However, only TP, $NO_3^-$-N, and $NH_4^+$-N showed highly significant positive correlations, with $R^2$ values of 0.715, 0.976, and 0.844, respectively.

**Figure 1.** Soil nutrient and correlation analysis. (**A**) Variation in soil nutrient content of different RNE treatments. Blue represents 0–20 cm soil. Red represents 20–40 cm soil. Black represents 0–40 cm soil. Fitting was performed using ordinary least square. (**B**) Correlation between different nutrients. Blue represents positive correlation. Red represents negative correlations. The size of the circles represents the level of correlation. Correlations were calculated using Pearson's coefficient. (**C**) Grey relational analysis of different nutrients with RNE. A redder color represents higher correlation.

## 3.2. Plant Yield Components and Correlation Analysis

With the RNE treatment, the changing pattern of seed cotton and lint yield were similar; both showed an increasing trend with the decrease of N fertilizer and then decreased (Figure 2A). The highest yield was observed in N2 treatment, with an average yield of 6354.33 kg/hm$^2$ (seed cotton) and 2664.12 kg/hm$^2$ (lint cotton) in both years, representing a 13.88%, 19.84%, 39.44% (seed cotton) and 18.1%, 24.98%, 41.28% (lint cotton) increase, respectively, compared to CK, N4, and N0 treatments. Further analysis of the yield components of each treatment showed that N reduction did not significant affect the number of harvested plants (Supplementary Table S1). However, both boll number and boll weight increased and then decreased with the reduction of N fertilizer in both years, with the highest values observed in the 20% N reduction treatment. The two-year mean values of boll number in N2 treatment increased by 6.54%, 14.37%, and 30.16%, respectively, compared to CK, N4, and N0 treatments. The single boll weight increased by 4.30%, 9.29%, and 13.62%, respectively. Moreover, the treatments showed significant difference ($p < 0.05$) among them. Cotton lint percent was highest in both years in N2 treatment, with an average lint percent of 41.92%, which was 3.48%, 9.29%, and 13.62% higher than that of CK, N4, and N0 treatments, respectively. These results indicated that

N reduction mainly affected cotton boll number, boll weight, lint percent, and ultimately the final yield. The high yield and N fertilizer values in 2020 were between N2 and N4 treatments, while in 2021, they were between CK and N2 treatments. Although there was a slight difference between the high values in the two years, it also suggests that current conventional N fertilizer use has the potential to be reduced while maintaining a stable plant yield.

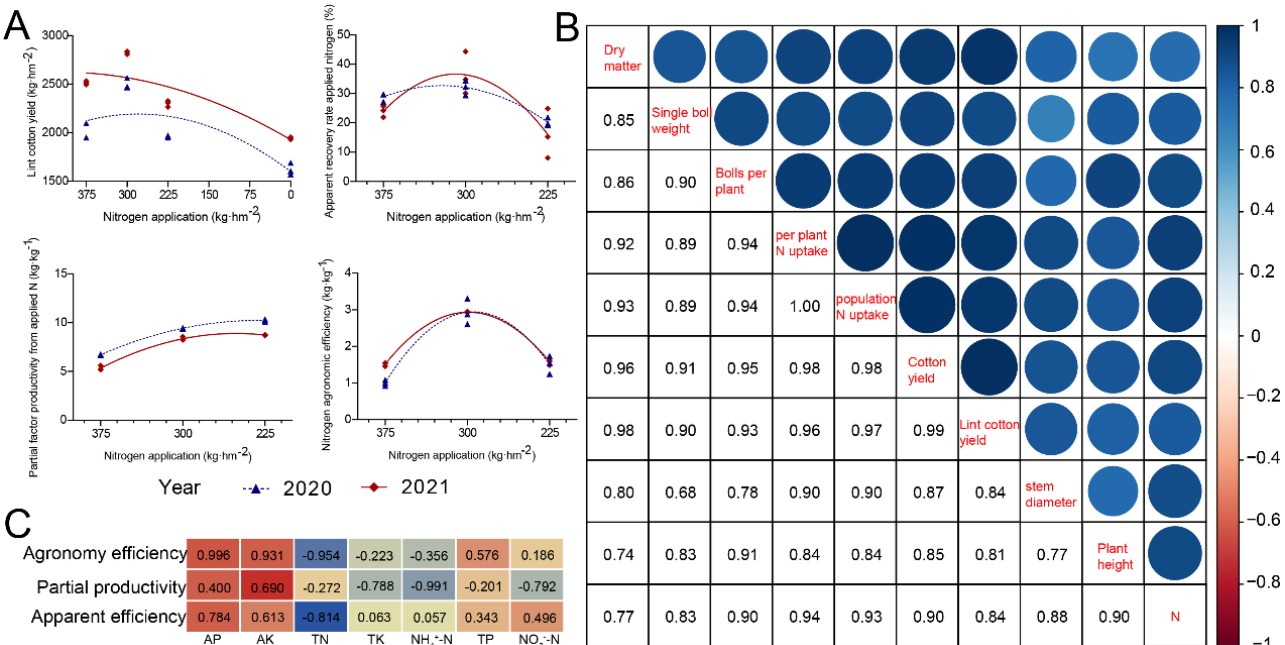

**Figure 2.** Cotton yield, correlation analysis and nitrogen use efficiency. (**A**) Lint yield and nitrogen use efficiency of different RNE treatments in two years. Blue represents the year 2020. Red represents 2021. Fitting was performed using ordinary least square. (**B**) Correlation analysis of yield components and RNE. Blue represents positive correlation. Red represents negative correlation. The size of the circles represents the level of correlation. Correlations were calculated using Pearson's coefficient. (**C**) Correlation analysis between nitrogen use efficiency and soil nutrients. Redder colors represent higher correlation. Correlations were calculated using Pearson's coefficient.

The correlations between N reduction and plant height, stem diameter, dry matter, nitrogen uptake, population nitrogen uptake, bolls per plant, single boll weight, cotton yield, and lint yield showed highly significant positive correlations (Figure 2B). The correlation coefficients ($R^2$) range from 0.656 to 0.964. The correlations among the indicators were further analyzed and compared. Cotton yield, lint yield, dry matter, per plant N uptake, population N uptake, bolls per plant, and single boll weight of cotton showed significant positive correlations with $R^2$ ranging from 0.739 to 0.985 in both years. Single boll weight, dry matter, per plant N uptake, population N uptake, and bolls per plant had highly significant positive correlations with $R^2$, ranging from 0.790–1.000, and all were positively correlated with plant height and stem diameter. These results indicate that N reduction is closely related to agronomic traits, material accumulation, and yield formation in cotton, and although there were differences in the correlations between each index and N reduction, the regulation of plant traits after N reduction was more reflective of multiple indexes rather than the regulation of a single index.

### 3.3. Grey Relational Analysis and Nitrogen Use Efficiency

In the 0–20 cm soil layer, the correlation analysis showed that $NH_4^+$-N had the highest correlation with N reduction, followed by AK, organic matter, TP, TK, AP, pH, $NO_3^-$-N, and TN, (Figure 1C). Similarly, in the 20–40 cm soil layer, $NO_3^-$-N had the highest correlation

with N reduction, followed by $NH_4^+$-N, AK, TP, pH, TK, AP, organic matter, and TN, with correlation coefficients ranging from 0.461 to 0.689. These results indicated that $NO_3^-$-N and AK were the most responsive soil indicators to RNE in both soil layers, while TN was not sensitive to RNE.

The nitrogen agronomic efficiency of cotton was negatively correlated with TN, TK, and $NH_4^+$-N (Figure 2C). The average correlation coefficient value was −0.943 for both years. However, the nitrogen agronomic efficiency was significantly positively correlated with AP and AK, with a correlation coefficient ranging from 0.920 to 0.966. The partial factor productivity from applied N was negatively correlated with soil TN, TP, TK, $NO_3^-$-N, and $NH_4^+$-N, particularly with TK, $NO_3^-$-N and $NH_4^+$-N. The apparent recovery rate of applied nitrogen was negatively correlated with TN, TP, TK, $NO_3^-$-N, and $NH_4^+$-N, and positively correlated with TP, TK, $NO_3^-$-N, $NH_4^+$-N, AP, and AK.

### 3.4. Identification of Soil Bacteria and Fungi in Different Soil Layers under RNE

To comprehensively explore the succession of soil microbial community structure in different soil layers under reduced N fertilizer, we identified bacteria and fungi in 40 soil samples subjected to different treatments by sequencing the conserved regions of microbial 16 rDNA v3 + v4 region and 18s rDNA ITs. We obtained a total of 3,173,460 filtered bacterial sequences (Supplementary Table S2), with 3,130,218 effective sequences, an average sequence length of 421 bp, an average GC content of 57.30%, and a lowest Q30 of 96.22% for CKA1. Similarly, we obtained 3,184,945 filtered fungal sequences, with 3,112,535 effective sequences, an average sequence length of 248 bp, an average GC content of 48.11%, and a lowest Q30 value of 98.85% for N0B3. These results indicated that the identification bacteria and fungi in different soil samples was accurate and reliable.

OTU clustering was performed based on the 97% similarity principle, with each OTU representing a class of microorganisms for both bacteria and fungi. The number of OTUs was counted for all samples, and the results showed that N0B had the highest number of bacteria with 1871 OTUs, while N2A had the lowest number of bacteria with 1836 OTUs, and an average of 1859 OTUs per soil sample (Figure 3A,B). ANOVA analysis showed no significant difference in the number of bacterial and fungal OTUs among all soil samples, indicating that there was no drastic change in soil microbial abundance under different soil layers or RNE at the OTU level.

Intersection and concatenation analysis of clustered OTU species and petal plots revealed (Figure 3C,D) that all treatment groupings shared 1803 bacterial OTUs and 1026 fungal OTUs, with no treatment-specific OTUs. These results further demonstrated that bacteria and fungi were more similar in soil samples from different treatments at the OTU level.

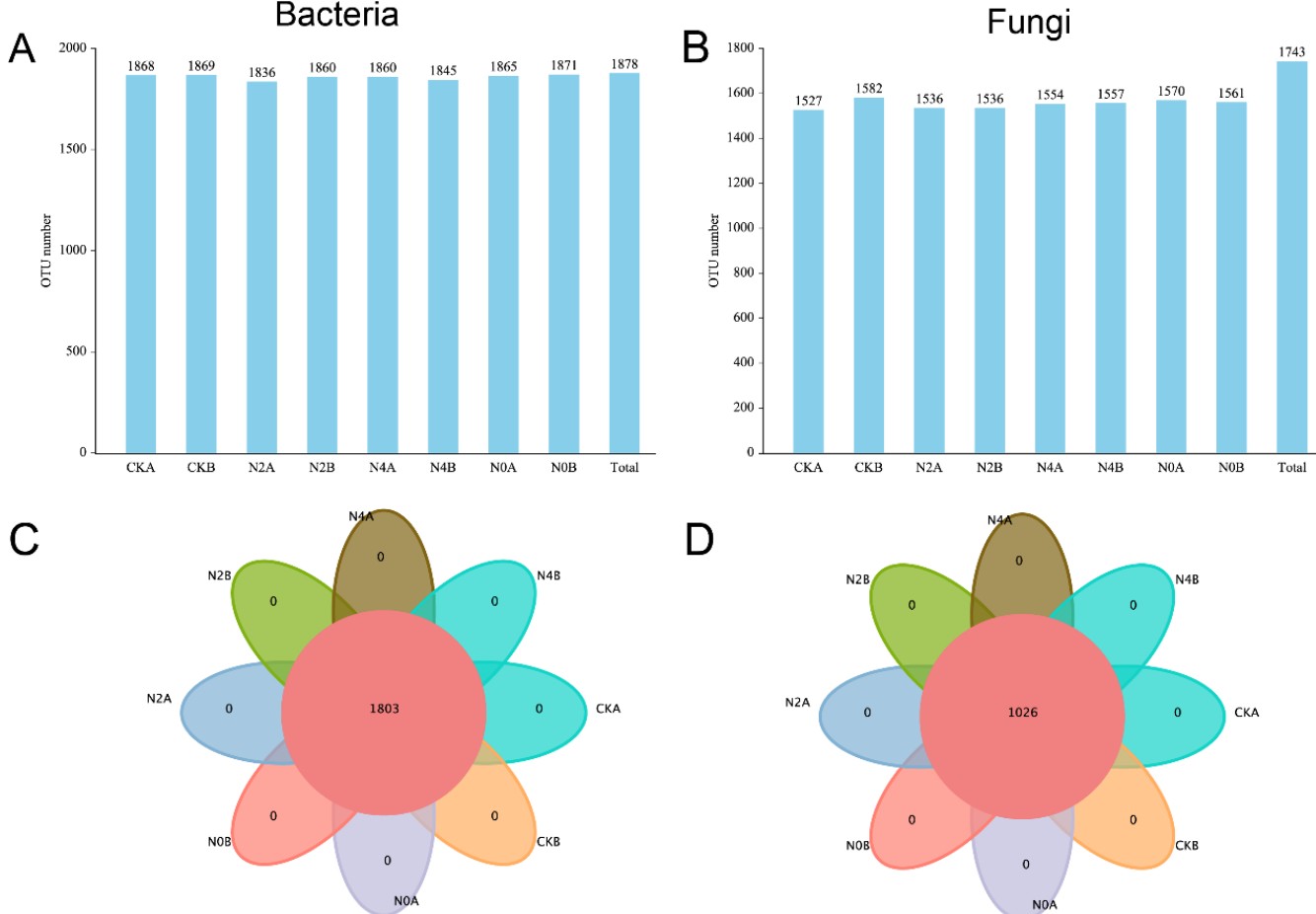

**Figure 3.** OTU statistics. (**A**) Histogram of bacteria OTUs number in different treatments. (**B**) Histogram of fungi OTUs number in different treatments. (**C**) Venn diagram of bacteria OTUs with different treatments. (**D**) Venn diagram of fungi OTUs with different treatments.

### 3.5. Analysis of Microbial Community Diversity and Abundance

The alpha diversity of the soil microbial community was measured using the ACE, Chao1, Simpson, and Shannon indices, where the ACE and Chao1 indices were positively correlated with microbial abundance, and the Simpson and Shannon indices were positively correlated with microbial diversity (Table 1). Bacterial diversity and abundance significantly decreased after a 20% N reduction in the 0–20 cm soil layer, and all four alpha diversity indices in the N2 group under this soil layer were significantly lower than those in the other groups. In the 20–40 cm soil layer under N fertilizer reduction, there was no significant change in bacterial diversity and richness, and all four $\alpha$-diversity indicators in the four groups of samples under this soil layer were not different. The fungal Chao1 index decreased significantly in N2A, and the abundance of fungi in the N2 soil under this soil layer decreased. The fungal ACE index increased significantly in N4B, and the fungal abundance in N4 samples under this soil layer was significantly higher than that in other subgroups. Overall, the microbial changes in the soils of each subgroup were small, and only the treatment with 20% RNE significantly changed the distribution of bacterial and fungal communities in the 0–20 cm soil layer, while the treatment with 40% RNE increased the abundance in the 20–40 cm soil layer. The diversity of bacteria and fungi in the deep soil layer was unchanged, which was consistent with the highly overlapping OTU results of the treatments, indicating that the soil microbial community structure was less changed in the treatments except for the 20% RNE. The coverage of all subgroup samples was more than

99%, indicating that these data adequately represented the microbial community structure under different treatments and soil layers with high reproducibility.

**Table 1.** Alpha diversity analysis of different samples.

| Category | Depth (cm) | Sample | ACE | Chao1 | Simpson | Shannon | Coverage |
|---|---|---|---|---|---|---|---|
| Bacteria | 0–20 | CK | 1789.56 ± 11.80 [a] | 1808.45 ± 12.36 [a] | 0.9965 ± 0.00007 [a] | 9.2153 ± 0.0191 [a] | 0.9986 |
| | | N0 | 1772.00 ± 5.34 [ab] | 1785.72 ± 4.89 [ab] | 0.9964 ± 0.00005 [ab] | 9.1571 ± 0.0194 [ab] | 0.9986 |
| | | N2 | 1725.51 ± 5.10 [b] | 1740.90 ± 8.02 [b] | 0.9959 ± 0.00011 [b] | 9.0625 ± 0.0316 [b] | 0.9986 |
| | | N4 | 1787.51 ± 6.00 [a] | 1801.99 ± 9.09 [a] | 0.9964 ± 0.00010 [a] | 9.231 ± 0.0303 [a] | 0.9987 |
| | 20–40 | CK | 1796.32 ± 9.70 [a] | 1801.98 ± 7.62 [a] | 0.9968 ± 0.00007 [a] | 9.30208 ± 0.0213 [a] | 0.9988 |
| | | N0 | 1713.39 ± 46.74 [a] | 1720.80 ± 53.05 [a] | 0.9921 ± 0.00435 [a] | 8.72612 ± 0.3914 [a] | 0.9971 |
| | | N2 | 1796.58 ± 5.88 [a] | 1809.07 ± 6.89 [a] | 0.9965 ± 0.00011 [a] | 9.2138 ± 0.0246 [a] | 0.9987 |
| | | N4 | 1771.34 ± 6.00 [a] | 1785.09 ± 9.78 [a] | 0.9966 ± 0.00005 [a] | 9.20336 ± 0.0133 [a] | 0.9986 |
| Fungi | 0–20 | CK | 1115.32 ± 45.34 [a] | 1182.79 ± 44.33 [ab] | 0.9916 ± 0.00017 [a] | 8.2311 ± 0.0355 [a] | 0.9987 |
| | | N0 | 1112.21 ± 60.46 [a] | 1230.61 ± 47.83 [a] | 0.9914 ± 0.00013 [a] | 8.2144 ± 0.0097 [a] | 0.9986 |
| | | N2 | 954.14 ± 14.19 [a] | 1024.35 ± 28.64 [b] | 0.9859 ± 0.00533 [a] | 8.0255 ± 0.1694 [a] | 0.999 |
| | | N4 | 1048.67 ± 64.67 [a] | 1116.14 ± 50.66 [ab] | 0.9905 ± 0.00065 [a] | 8.1488 ± 0.0610 [a] | 0.9988 |
| | 20–40 | CK | 1046.40 ± 34.43 [b] | 1144.51 ± 43.98 [a] | 0.9904 ± 0.00097 [a] | 8.1777 ± 0.0520 [a] | 0.9988 |
| | | N0 | 1067.76 ± 39.73 [b] | 1146.86 ± 32.23 [a] | 0.9903 ± 0.00088 [a] | 8.1708 ± 0.0549 [a] | 0.9988 |
| | | N2 | 1033.32 ± 27.14 [b] | 1117.48 ± 22.75 [a] | 0.9906 ± 0.00108 [a] | 8.1500 ± 0.0881 [a] | 0.9988 |
| | | N4 | 1260.62 ± 23.60 [a] | 1239.77 ± 30.63 [a] | 0.9904 ± 0.00048 [a] | 8.1757 ± 0.0359 [a] | 0.9982 |

Note. Significance is marked with lowercase letters. *p*-value < 0.01.

### 3.6. Analysis of Microbial Community Structure of Soils

To assess differences in soil microbial community structure in different soil layers under RNE, beta analysis of bacteria and fungi was performed at the OTU level using binary Jaccard and Bray–Curtis algorithms, respectively. The results of the PCA principal component analysis showed (Figure 4A) that the significant PCAs in bacteria explained minor differences in community structure, with PC1 explaining 26.09% and PC2 explaining 21.34%. However, the loadings plots showed that different samples were distributed in different matrix quadrants (Figure 4B). The same group of samples was closer in spatial distance, and the samples of different treatments showed significant separation at a lower explanation degree, indicating that the RNE significantly changed the community structure of bacteria in different soil layers. Samples from the CK were located on the left side of the matrix quadrant, while the RNE treatments were located on the right side, indicating that PC1 was correlated with the RNE treatment. Except for the N0 treatment with 100% N reduction, all samples from the 0–20 cm soil layer were located above the matrix quadrant, and the 20–40 cm soil layer samples were located below it, indicating that PC2 was correlated with the soil layer depth. These findings were consistent with the results of the correlation heat map (Figure 4D), which showed that PC1 had a 71% correlation with the RNE treatment and PC2 had a 72% correlation with soil depth. The gravel plot provided further results (Figure 4C), showing that OTU6 had the highest positive correlation and OTU83 had the highest negative correlation with PC1, suggesting that both OTUs may be correlated with the RNE treatment. Additionally, OTU18 had the highest positive correlation and OTU15 had the highest negative correlation with PC2, indicating that both OTUs may be correlated with soil depth. The multivariate ANOVA revealed differences within treatments (Figure 5A); the significant difference in bacterial community structure occurred between 20% and 40% N reduction in the 0–20 cm soil layer, and between 40% and 100% N reduction under the 20–40 cm soil layer.

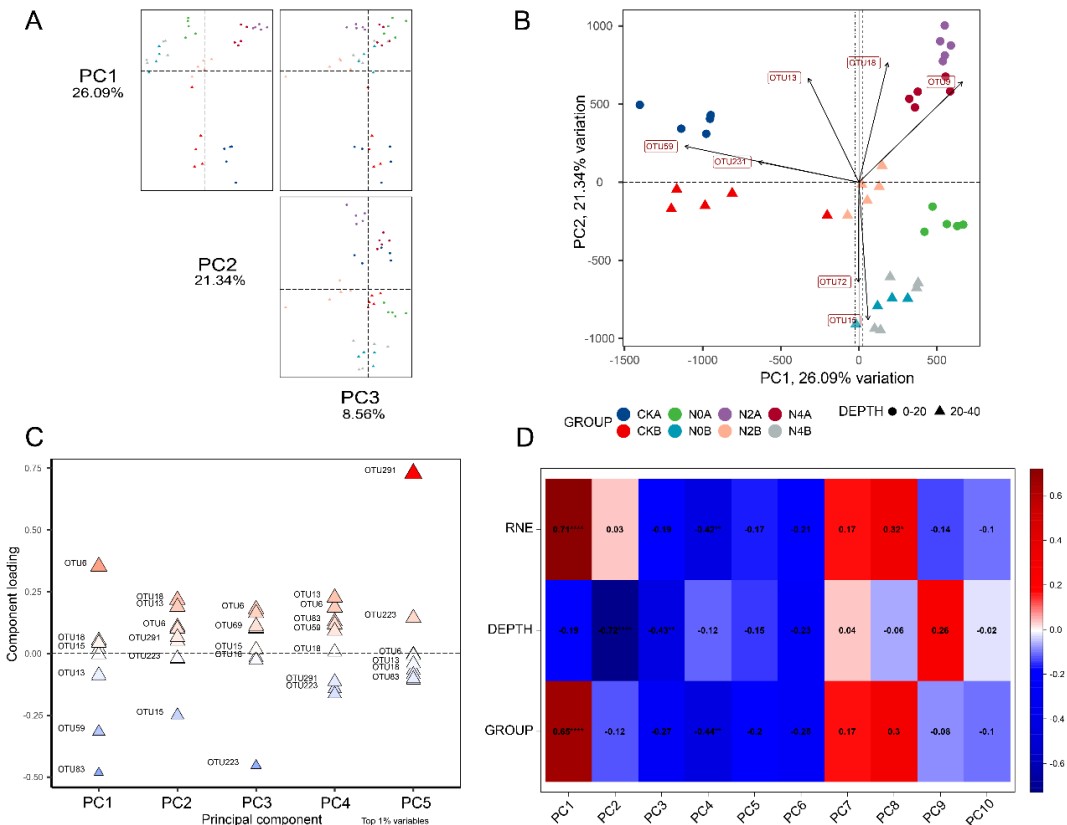

**Figure 4.** Bacterial PCA analysis. (**A**) Score plot. The first three PCs were shown. The percentages represent the variance explanation scale. The points represent the observation sample. Treatments are shown in different colors. (**B**) Biplot. Dots represent observation samples. Circles are 0–20 cm soil. Triangles are 20–40 cm soil. RNE treatments are shown in different colors. The line segments represent the main differences between OTUs. The magnitude of the angle between the line segments and the points represents the positive, negative and level correlation and the level. (**C**) Loadings plot. The OTUs with a significant relationship with PC1–5 are shown. Redder colors mean higher correlation. (**D**) Eigencor plot. Correlations between PC1–10 and RNE, soil depth and sample grouping were demonstrated. Red represents positive correlations and blue represents negative correlations. Darker colors represent higher correlations, * represents the significance to $1 \times 10^{-2}$, ** represents significance to $1 \times 10^{-3}$ and **** represents $1 \times 10^{-5}$.

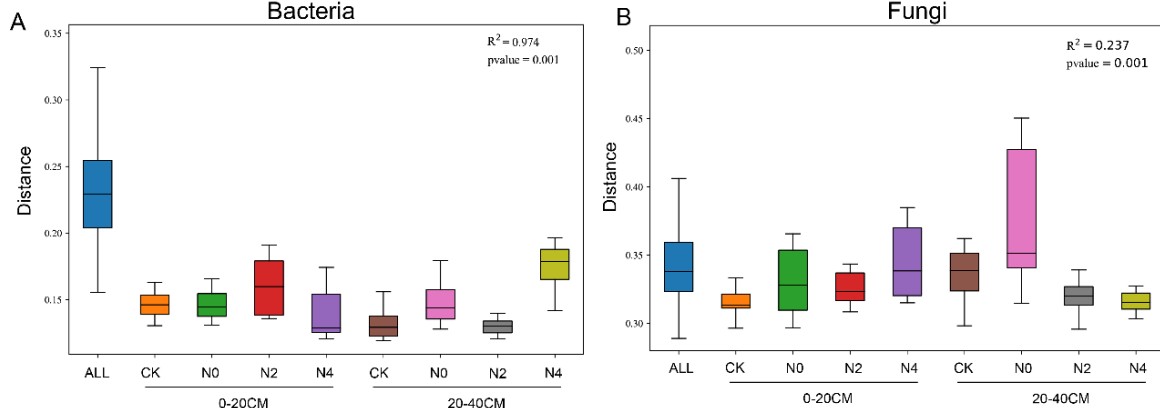

**Figure 5.** Analysis of variance in bacteria and fungi. The horizontal axis represents different samples. The vertical axis represents the distance. The horizontal line in the box represents the average. (**A**) Bacteria. (**B**) Fungi.

The PCA analysis of fungi OTUs revealed that the degree explanation of PC1 and PC2 was low, with PC1 explaining 18.46% and PC2 explaining 12.04 (Figure 6A). In contrast to the bacterial PCA analysis, the community structure of fungi within each treatment was significantly discrete, with samples of the same treatment distributed in different matrix quadrants and spatially distant from each other (Figure 6B). Outliers were observed in the CKB treatment, N2B treatment, and N4B treatment, indicating that the effect of RNE on the fungal community structure of the soil in different soil layers was limited. The fungal community structure varied greatly among different samples. The results of the correlation heatmap showed that PC1 had a negative correlation of 38% with soil depth, and PC2 had a negative correlation of 30% with soil depth (Figure 6D). They were almost entirely uncorrelated with the RNE treatment, indicating that the differences in the structure of fungal communities were mainly caused by soil depth. The multivariate ANOVA revealed (Figure 5B) that although the difference values of most treatments were smaller than the difference between groups, indicating that RNE caused a change in fungal community structure, the difference values of N4A and N0A were still more extensive than the difference between groups. The overall dispersion of fungal OTUs was large.

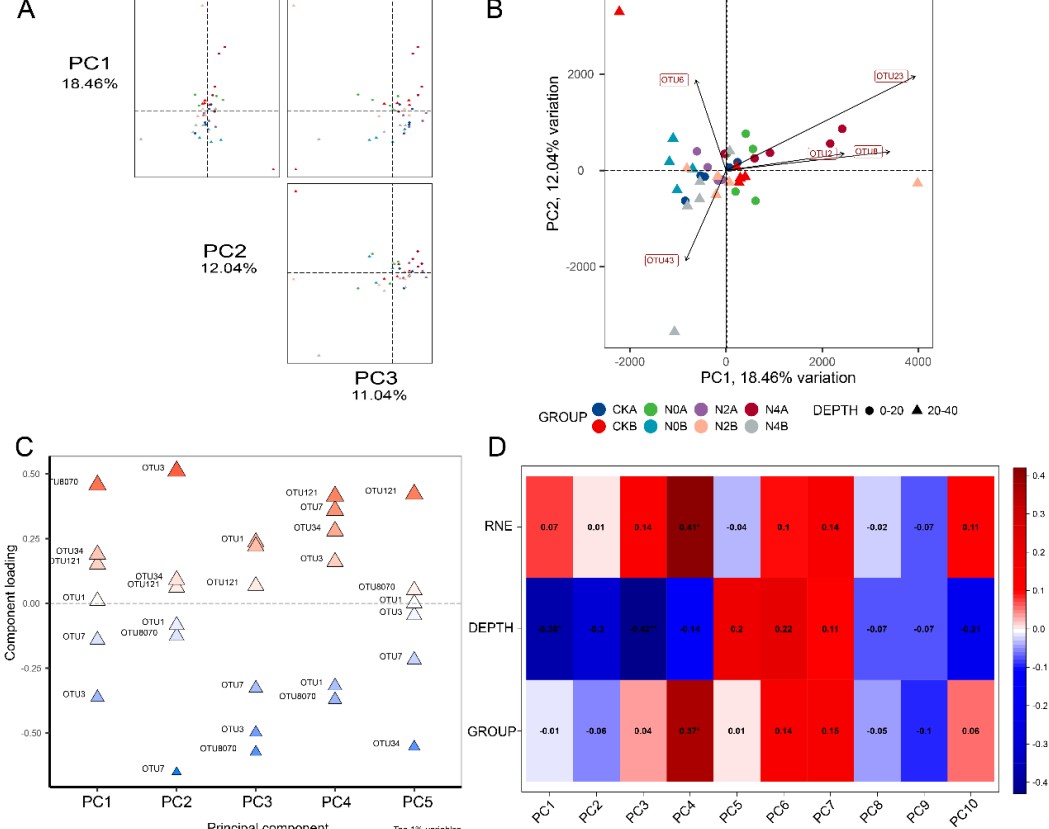

**Figure 6.** Fungi PCA analysis. (**A**) Score plot. The first three PCs were shown. The percentages represent the variance explanation scale. The points represent the observation sample. Treatments are shown in different colors. (**B**) Biplot. Dots represent observation samples. Circles are 0–20 cm soil. Triangles are 20–40 cm soil. RNE treatments are shown in different colors. The line segments represent the main differences OTUs. The magnitude of the angle between the line segments and the points represents the positive, negative correlation and the level. (**C**) Loadings plot. The OTUs with significant relationship with PC1–5 are shown. Redder colors represents a higher correlation. (**D**) Eigencor plot. Correlations between PC1–10 and RNE, soil depth and sample grouping were demonstrated. Red represents positive correlations and blue represents negative correlations. Darker colors represent higher correlations, * represents significance to $1 \times 10^{-2}$ and ** represents $1 \times 10^{-3}$.

### 3.7. Bacterial Species Composition of Soils

All bacterial OTUs were annotated using the SILVA public database, and microbial species with an abundance percentage greater than 1% were retained. A total of 18 phyla, 50 orders, 96 orders, 115 families, 134 genera, and 138 species were detected in bacteria. At the phylum taxonomic level, Proteobacteria, Acidobacteria, and Gemmatimonadetes were the dominant phyla, accounting for 31.76–36.99%, 25.13–30.21%, and 9.12–11.33% of all samples, respectively (Figure 7A). The classification statistics of the top ten phyla in terms of abundance ratio showed that the abundance of Acidobacteria increased with RNE under different soil layers. The abundance percentage at 20% N reduction was 29.91% and 27.39%, respectively, compared to 25.89% and 25.23% in the control, which represents a 4.02% and 2.16% increase, respectively. In contrast, the abundance percentages of both Bacteroidetes and Firmicutes decreased with the RNE under different soil layers. Specifically, the abundance percentages of Bacteroidetes in the N2 treatment were 3.42% and 3.21%, respectively, which were 1.06% and 1.07% lower than in the control. The abundance proportions of the Firmicutes were 0.96% and 1.10% in N2 treatment, which were 3.05% and 1.91% lower than in the CK, respectively. The remaining bacterial abundances did not change significantly.

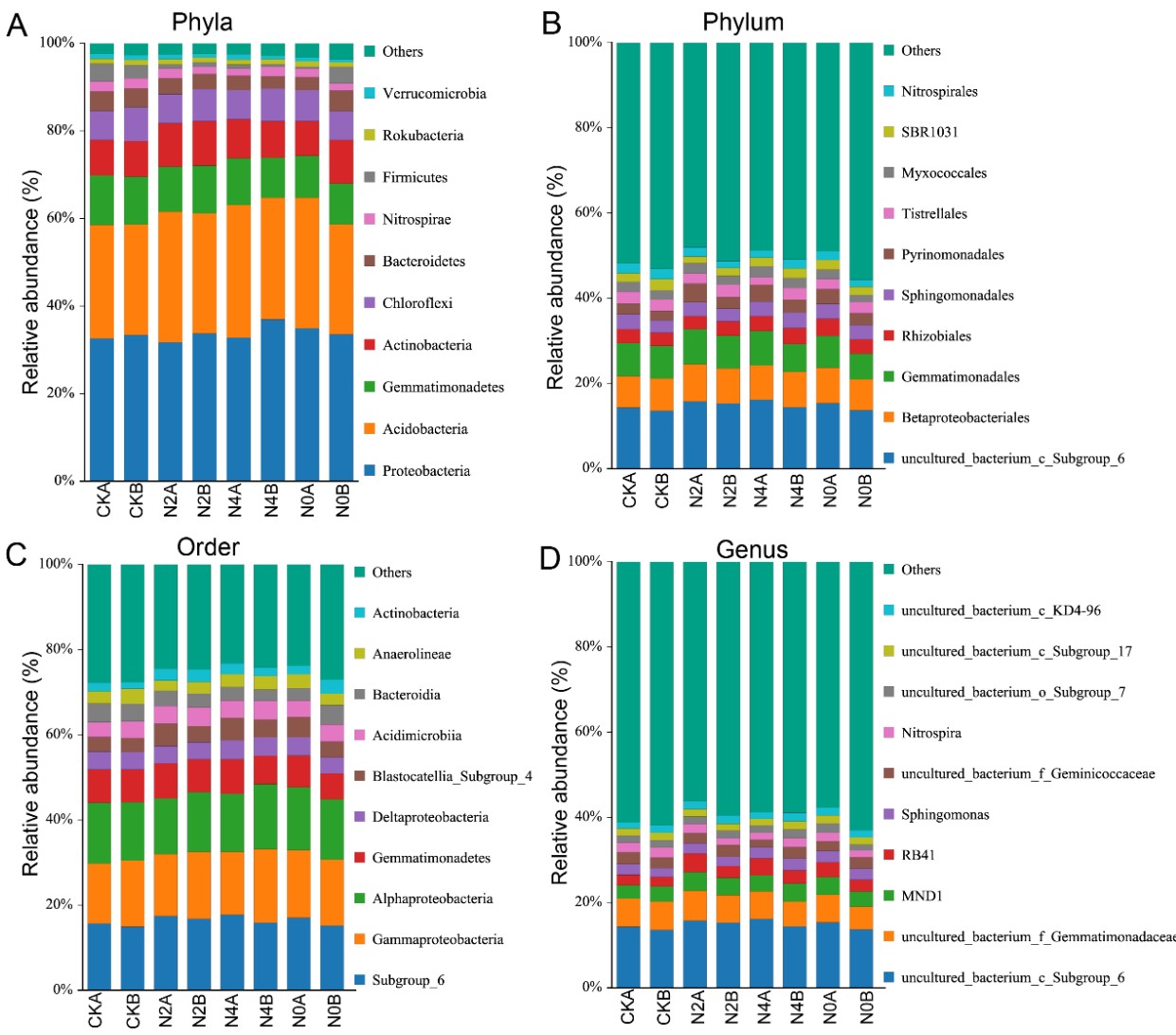

**Figure 7.** Relative abundance of dominant bacteria at different taxonomic levels. The horizontal axis represents different samples. The vertical axis represents the relative abundance. (**A**) Phyla level. (**B**) Phylum level. (**C**) Order level. (**D**) Genus level.

At the phylum level, Gammaproteobacteria and Alphaproteobacteria were the dominant phyla, accounting for 14.07–17.33% and 13.10–15.28% of all samples, respectively (Figure 7B). The top ten phyla in terms of abundance ratio showed that the relative abundance of both Gammaproteobacteria and Alphaproteobacteria decreased with the increasing RNE. The relative abundance of Gammaproteobacteria was highest in N0A treatment, while the relative abundance of Alphaproteobacteria reached its maximumin the N4B, with a 1.52% increase compared to the CK. In contrast, the relative abundance of Bacteroides decreased under RNE, with 3.55% and 3.19% in different soil layers under 20% nitrogen reduction, representing a decrease of 0.81% and 0.92%, respectively, compared to the CK.

At the order level (Figure 7C), Betaproteobacteriales and Gemmatimonadales were found to be the dominant orders, accounting for 7.23–8.77% and 6.03–8.18% of all samples, respectively. The analysis of the top ten bacterial orders in terms of abundance revealed significant variation in the bacterial orders in different samples; RNE caused an increase in the relative abundance levels of Betaproteobacteriales and Rhizobiales, with the highest relative abundance of Bacteroides in the R2A. The highest abundance of Betaproteobacteriales was found in the N2A, reaching 8.77%, which increased by 1.37% compared with the CK in the same soil layer. The highest percentage of Rhizobiales abundance was observed in the N0A, at 4.04%, which increased by 0.85% compared with the CKA. In contrast, the relative abundance of SBR1031 and Nitrospirales decreased with RNE. SBR1031 was observed to be the lowest in the N2A, with a relative abundance of 1.58%, a decrease of 0.45% compared to the control. Similarly, Nitrospirales was the lowest in the N2B, with a relative abundance of 1.62%, a decrease of 0.76% compared to the CK. No significant changes were observed in the rest of the fungal orders.

At the genus level (Figure 7D), MND1 and RB41 were the dominant genera, with MND1 accounting for 3.17–4.38% of all samples and RB41 accounting for 2.22–4.39% of all samples. Taxonomic statistics of the top ten genera in terms of abundance showed that among the successfully annotated genera, the abundance proportions of MND1, RB41 and Sphingomonas increased with RNE. The highest relative abundance of MND1 and RB41 was observed in the N2A treatment, at 4.38% and 4.39%, respectively, which increased by 1.21% and 1.96% compared to the CKA. The highest relative abundance of Sphingomonas was 2.77% in the N4B treatment, which was 0.72% higher than the CK. In the 20–40 cm soil layer, the abundance of Dongia decreased with RNE, and the lowest relative abundance of this bacterium was 1.17% in the N0B treatment, which was 0.3% lower than the CK.

### 3.8. Fungal Species Composition of Soils

Fungal OTUs were annotated using the UNITE public database, and species with an abundance ratio greater than 1% were identified. A total of 11 phyla, 23 orders, 47 orders, 76 families, 107 genera, and 80 species were detected in fungi. Ascomycota and Basidiomycota were the dominant phyla, accounting for 69.20–72.71% and 15.05–16.41% of all samples, respectively (Figure 8A). Among the top ten ranked phyla, the abundance of Mortierellomycota under the 20–40 cm soil layer increased with RNE, reaching a highest relative abundance of 8.10% under N4, which was 2.20% higher than the CK. Additionally, the abundance of Chytridiomycota increased with the RNE in different soil layers. The abundance of Chytridiomycota in the CK was 0.85% and 0.77%, respectively. Meanwhile, in the N4 treatment, it was 1.46% and 0.95%, respectively, which increased by 0.61% and 0.18% compared to the CK. The remaining phyla did not show significant changes in abundance.

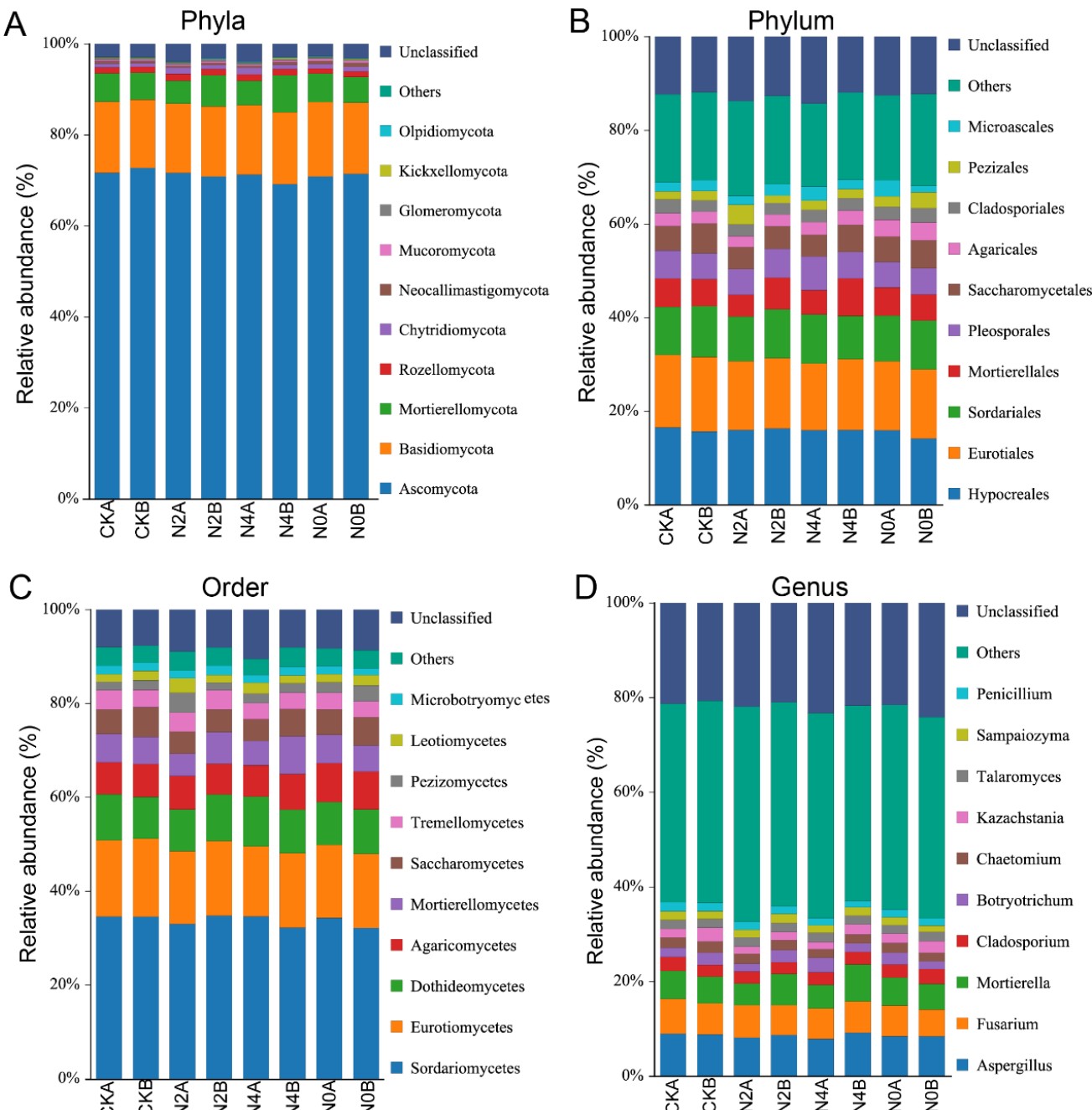

**Figure 8.** Relative abundance of dominant fungi at different taxonomic levels. The horizontal axis represents different samples. The vertical axis represents the relative abundance. (**A**) Phyla level. (**B**) Phylum level. (**C**) Order level. (**D**) Genus level.

At the Eumycetes level (Figure 8B), the dominant classes were Sordariomycetes and Eurotiomycetes, accounting for 32.12–34.79% and 14.92–16.62% of all samples, respectively. The relative abundance of Saccharomycetes decreased with the RNE, and the highest abundance of Saccharomycetes was observed at R4A, at 4.65% abundance, which was 0.57% higher than the CK. The relative abundance of Pezizomycetes showed opposite trends in different soil layers. The relative abundance of Pezizomycetes in the 0–20 cm soil layer was positively correlated with RNE, reaching the highest at N2, with a 2.50% increase compared to the CK. In contrast, the relative abundance of Pezizomycetes in the 20–40 cm soil layer was negatively correlated with RNE, and the lowest abundance was observed at

N2, which decreased by 0.44% compared to the CK. Similarly, the relative abundance of Pezizomycetes in the lower disc of the 20–40 cm soil layer was also negatively correlated with the RNE, with the lowest relative abundance observed at N2, which was 0.44% lower than the CK. The remaining Eumycetes did not show a significant change.

At the order level, Hypocreales and Eurotiales were identified as the dominant orders, accounting for 14.19–16.55% and 14.22–15.92% of all samples, respectively (Figure 8C). Taxonomic analysis of the order revealed that the RNE led to an increase in the relative abundance of Pezizales, with the highest abundance of 4.21% observed in the N2A soil layer, representing a 2.50% increase compared to the CK in the same soil layer. On the other hand, nitrogen reduction treatment resulted in a decrease in the relative abundance of Eurotiales, with the lowest percentage observed in the N4A, accounting for 14.22% and indicating a 1.26% decrease compared to the CK. No significant changes were observed for the remaining orders. These findings suggest that the RNE had varying impacts on the different orders of fungi present in the soil.

At the genus level (Figure 8D), Aspergillus and Fusarium were the dominant genera, accounting for 7.91–9.22% and 5.67–7.37% of all samples, respectively. Analysis of the top ten genera showed that the relative abundance of Chaetomium and Penicillium decreased under the RNE. The lowest percentage of Chaetomium was 1.78% in the N4A soil layer, which was 0.41% lower than the CK in the same soil layer. The lowest abundance of Penicillium was 1.37% in the N4B soil layer, 0.50% lower than the CK. The relative abundance of fusarium in the 0–20 cm soil decreased with RNE. The lowest fusarium abundance was 0.68% the N2 soil layer, which was 0.54% lower than the CK in the same layer. No significant changes were observed for the remaining genera.

*3.9. Functional Prediction of Bacteria and Fungi in Soils*

To investigate the changes in microbial functions in response to RNE at different soil depths, the annotated bacterial functional composition was predicted based on FAPRO-TAX and the database. The results indicated that the main functional compositions in bacteria were chemoheterotrophy, aerobic_ammonia oxidation, nitrification, and aerobic_heterotrophy (Figure 9A). The nitrogen reduction treatment enhanced the functions of nitrification and aerobic_ammonia_oxidation, with the highest values observed in N2A, accounting for 21.29% and 15.72%, respectively, representing an increase of 4.10% and 4.20% compared to the CK. The functional composition of chemical heterotrophic and fermentation functions decreased, with the lowest values observed in N2A, representing a decrease of 2.70% and 4.08% compared to the CK. Additionally, the functional composition of nitrate_reduction significantly decreased in the N2A, with a relative abundance of 1.57%, representing a decrease of 0.24% compared to the CK. Functional differences between the N2 and CK were further analyzed based on the 95% functional confidence level and 1% significance level (Figure 9B). Our analysis showed that aerobic_ammonia_oxidation and nitrification were significantly reduced under the N2 treatment compared to the CK, while nitrification functions were significantly increased. The composition of chemoheterotrophy and fermentation functions was also significantly reduced.

The functional composition of soil fungi was predicted based on the FUNGuild database. The main functional compositions of soil fungi were identified as plant pathogen, animal pathogen, and wood saprotroph (Figure 9C). Minimal changes in the multifunctional composition of fungi were observed in the 20–40 cm soil layer, which is consistent with the alpha analysis results. However, in the 0–20 cm soil layer, The functional composition of phytopathogenic fungi increased under RNE. The highest proportion of phytopathogenic fungi was observed in the N2 treatment, with a functional proportion of 21.62%, an increase of 2.74% compared to the CK. On the other hand, the functional composition of animal pathogenic fungi decreased, with the minimum functional proportion of animal pathogenic fungi being 12.62% in the N4, which decreased by 0.94% compared to the CK. The functional composition of woody saprophytes also decreased, with the lowest functional proportion of 11.20% observed in the N2, which was 1.42% lower than the CK. Although the functional

composition of fungi in the 0–20 cm soil layer increased or decreased under N2 compared to the CK, none of the functional differences reached a significant level.

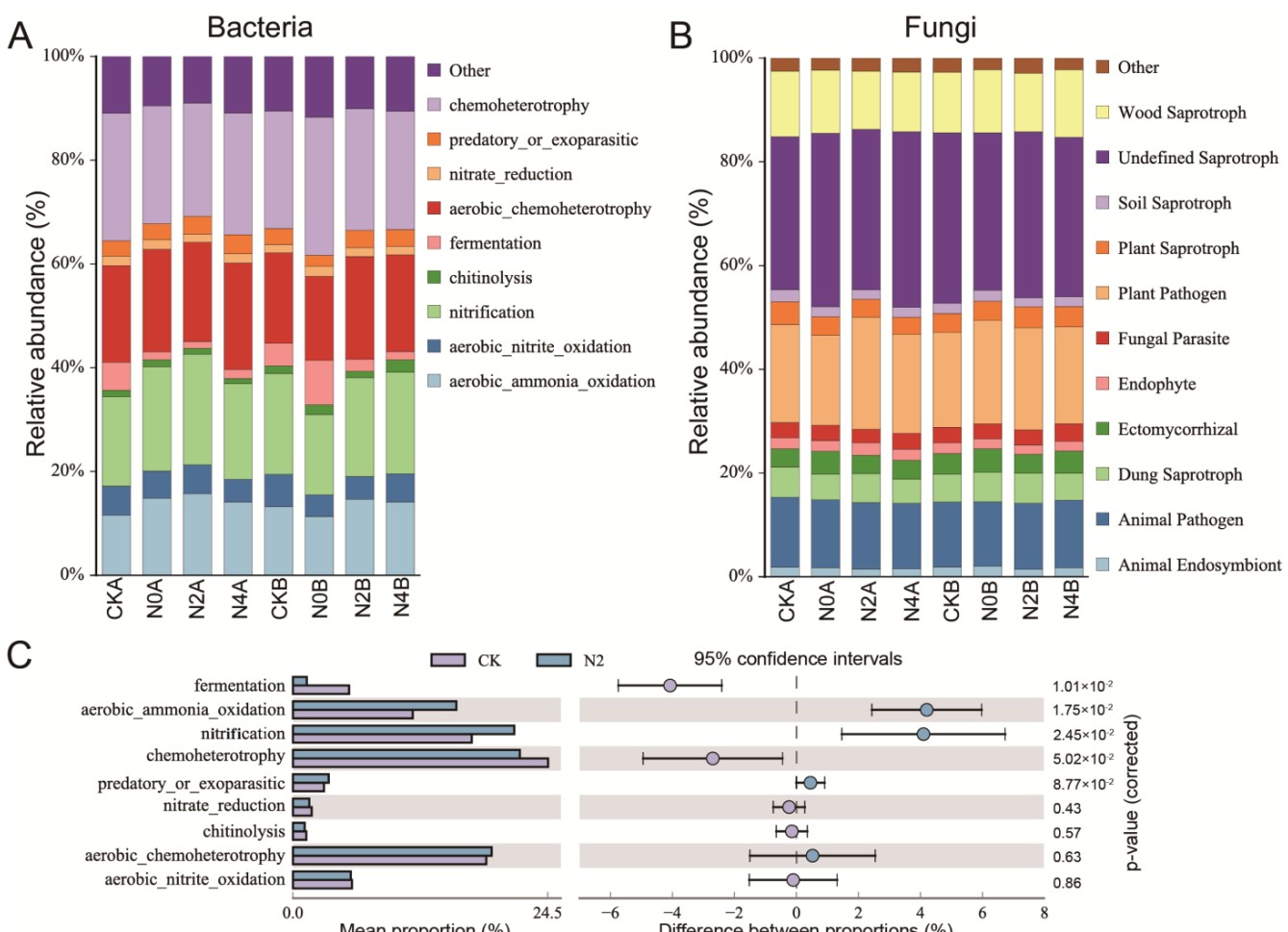

**Figure 9.** Functional analysis of bacteria and fungi. (**A**) The main function component of bacteria. The horizontal axis represents different samples. The vertical axis represents the relative abundance. (**B**) The main function component of fungi. The horizontal axis represents different samples. The vertical axis represents the relative abundance. (**C**) Significance test of functional differences between CK and N2 treatment. The bar chart on the left represents the difference in function. The blue bars are N2 treatments. The purple bars are CK. The length of the bars represents the relative abundance. The significance test of differences in 95% confidence intervals is shown on the right. The horizontal axis represents the fold change. The vertical axis represents the *p*-value.

*3.10. Correlation Analysis between Microorganisms and Soil Physicochemical Properties*

According to the Mantel test, the correlations between various physicochemical properties of the soil and the structures of bacterial and fungal community were analyzed (Table 2). The results showed that $NH_4^+$-N (R = 0.689, *p* = 0.001) and PH (R = 0.535, *p* = 0.001) were the main factors drove the variation in bacterial community structure in the soil. The correlations between soil physicochemical properties and the variation in fungal community structure were weak, and only AP (R = 0.219, *p* = 0.046) showed a weakly correlated with soil fungal community structure variation.

**Table 2.** Mantel test between soil nutrients and microorganisms.

| Category | Result | TN | NO$_3^-$-N | NH$_4^+$-N | AP | TP | TK | AK | OM | PH |
|---|---|---|---|---|---|---|---|---|---|---|
| Bacteria | R value | 0.129 | 0.496 | 0.689 | 0.198 | 0.088 | 0.396 | −0.082 | 0.304 | 0.535 |
| | *p*-value | 0.058 | 0.001 | 0.001 | 0.008 | 0.160 | 0.001 | 0.807 | 0.003 | 0.001 |
| Fungi | R value | −0.007 | −0.030 | −0.085 | 0.219 | −0.053 | 0.084 | −0.138 | 0.132 | −0.020 |
| | *p*-value | 0.434 | 0.681 | 0.695 | 0.046 | 0.624 | 0.183 | 0.853 | 0.214 | 0.450 |

The relationship between soil physicochemical properties and bacterial community structure was explored through redundancy analysis (RDA). RDA1 explained 16.62%, and RDA2 explained 13.97% of the total variation between bacterial community structure and soil physicochemical properties (Figure 10A). The samples from different treatments were located in different matrix quadrants. Both CK samples were located in the lower-left quadrant and were highly positively correlated with Nitrospira. The formation of the bacterial community structure in the CK was mainly driven by nitrate and ammonium nitrogen in the soil. Both sets of samples were positively correlated with the abundance of most major bacteria, negatively correlated with Nitrospira, and not correlated with Sphingomonas. AP was highly significant in driving bacterial community formation in the N2A and N4A, and TP, TK, and OM were also associated with variation in their community structure. Microorganisms that were positively associated with the N2 and N4 treatments mainly included Steroidobacter, RB41, Gemmatimonas, Ellin6067, Haliangium, and Sphingomonas. The samples with N0B and N4B treatment were distributed in the right matrix quadrant, and these three groups were negatively correlated with most bacterial abundances and positively correlated with Sphingomonas. PH and TN played a significant role in the structural variation of bacterial communities in these samples.

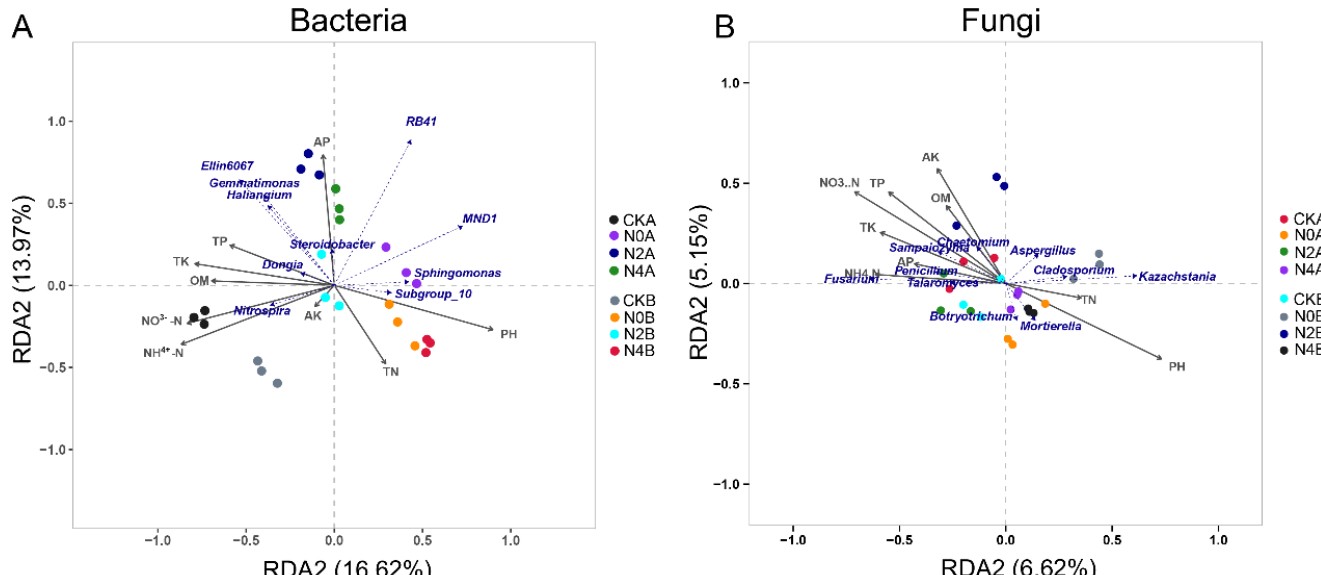

**Figure 10.** Redundancy analysis of bacteria and fungi. The significant difference quadrant is divided by gray dashed lines. Dots represent observed samples. Treatments are shown in different colors. Solid lines represent soil nutrients. The blue dashed lines represent the major differential microorganisms. The magnitude of the angle between the line segments and the points represents the positive, negative correlation and the level. (**A**) Bacteria. (**B**) Fungi.

Redundancy analysis was conducted to investigate the relationship between soil physicochemical properties and fungal community structure. The results revealed that RDA1 explained 6.62% and RDA2 explained 5.15% of the total variation in fungal community structure and soil physicochemical properties (Figure 10B). The soil samples from

N4 and N0 treatments were located in the right matrix quadrant, showing significantly positively correlations with Mortierella, Botryotrichum, Kazachstania, and Cladosporium. TN and PH were the main drivers of the formation of fungal community structure in the soil. The fungal community structure of the remaining treatments with lower or no RNE did not undergo significant hierarchical clustering or separation, and the spatial distances between different treatments were close to each other. The treatments with lower or no RNE had a lesser impact on the fungal community in the 0–40 cm layer. Fusarium, Talaromyces, Penicillium, Sampaiozyma, and Chaetomium were positively correlated with the fungal community structure of these samples.

### 3.11. Co-Occurrence Network of Bacterial and Fungal Communities

The interrelationships between bacterial genera in different soil layers in RNE treatments were explored using Spearman's correlation coefficient, and the co-occurrence network was analyzed. The bacterial co-occurrence network in the 0–20 cm soil layer consisted of 59 nodes and 200 edges, which were divided into eight clusters (Figure 11A). Cluster1 had the number of bacteria, with 13 genera, and the highest relative abundance of MND1, which was correlated with five different bacteria and was the core node of this cluster. Cluster 8 contained only one genus of Anaerolineae and SBR1031, with the lowest number of nodes. Cluster 6 contained 12 different genera, among which the relative abundance of Geminicoccaceae and Nitrospira were higher and correlated with 11 and 5 different bacteria, respectively, and were the core genera. Moreover, the bacterial genera in Cluster 1, 2, 3, 4 were directly correlated with Cluster 6, and these clusters were crucial for the construction of bacterial co-occurrence networks. Clusters 5 and 7 were related and independent of other clusters. Cluster 5 contained ten different genera, and Microscillaceae were the core nodes of this cluster due to their high abundance and co-occurrence with six different genera. Cluster 7 contained four genera, and both clusters were related to each other by Latescibacteria.

The co-occurrence network of bacterial genera in the 20–40 cm soil layer comprised 62 genera and 200 edges (Figure 11B). All nodes were classified into eight distinct clusters. which showed greater consistency in their interrelationships compared to the shallow soil layer. The primary six clusters were connected by SWB02, Aquicella, Bacteria, and Bacteroides, which played an important role in the stabilizing the bacterial community structure in the 20–40 cm soil layer. Cluster 2 comprised 17 different genera and had the highest relative abundance of Gemmatimonadaceae and Gemmatimonas, which interacted with a total of 17 different genera, occupying the core of the co-occurrence network of the cluster. Cluster 1 contained 12 different genera, among which the relative abundance of unclassified bacterial subgroup 6 was significantly higher than other nodes. Cluster 5 also contained 12 different nodes, with Sphingomonas and RB41 having a higher relative abundance, correlated with nine and four different genera, respectively, and being the core nodes of the cluster. MND1 was located at the core of the Cluster 3 network. The core nodes of Clusters 3 and 5 were close to each other in the co-occurrence network and had strong correlations. Interestingly, the core node of Cluster 5 was absent in the shallow soil layer's co-occurrence network. This result suggested that the nitrogen reduction treatment caused a variation in the bacterial community structure in the shallow soil layer, and part of the co-occurrence relationship was disrupted. In contrast, more correlations existed between bacteria in the deep soil layer.

The co-occurrence network analysis of fungal communities revealed interesting findings in both shallow and deep soil layers. In the shallow soil layer, the fungal cooccurrence network consisted of 70 different genera and 200 edges, and all genera were clustered into nine clusters based on correlation (Figure 12A). Among these clusters, Cluster 2 contained 15 genera, including Chaetomium and Kazachstania, which had the highest relative abundance and were correlated with five and four different genera, respectively, as the core nodes. Both Cluster 8 and Cluster 9 contained only two genera, with Cluster 8 consisting of Thelonectria and Gliocladiopsis, and Cluster 9 consisting of Llyonectria and Epicoccum.

Notably, Cluster 1 was related to multiple remaining clusters located at the core of the co-occurrence network, and Sampaiozyma, a fungal genus with a high relative abundance, was strongly correlated with two different genera.

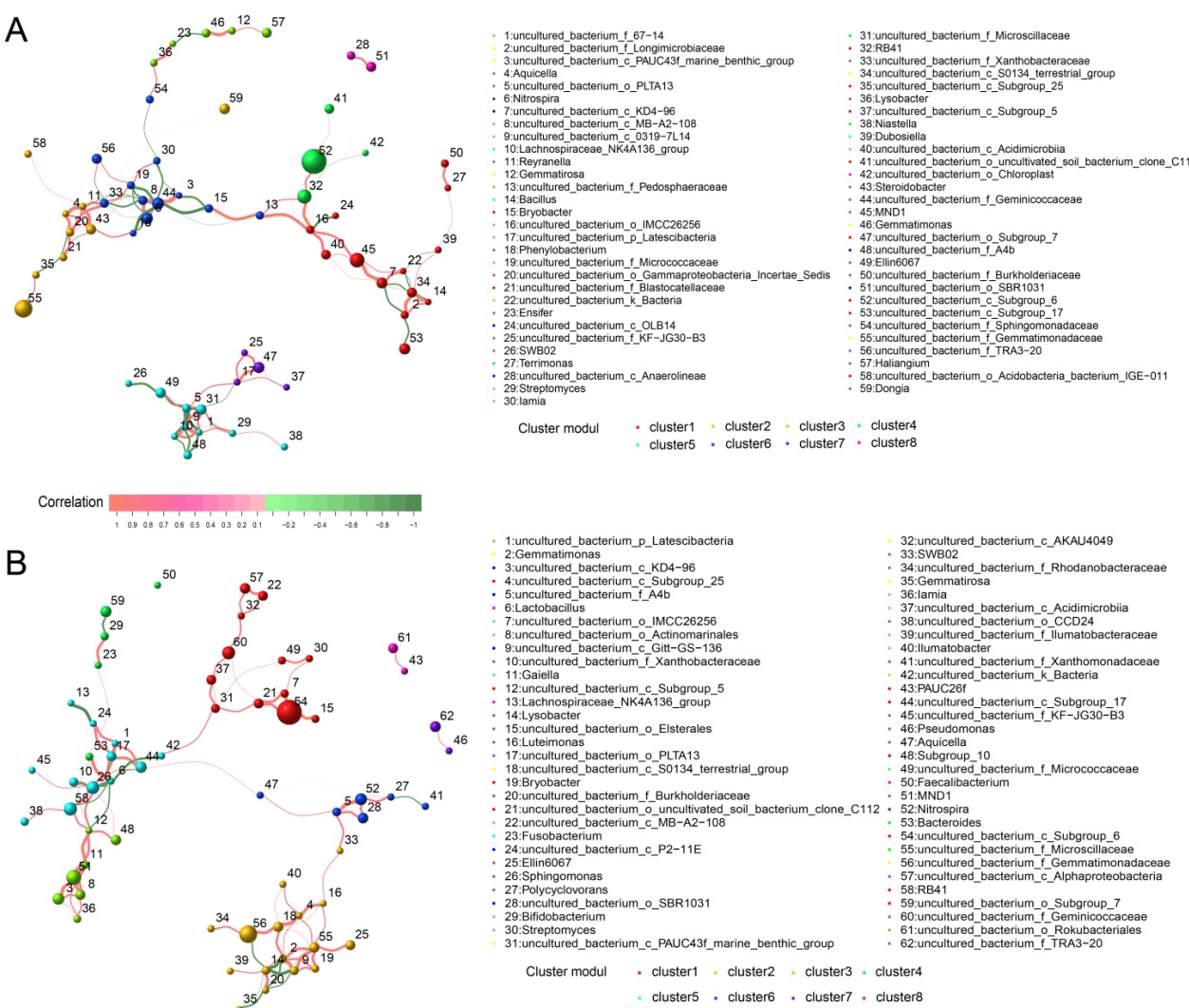

**Figure 11.** Co-occurrence network of bacteria. Each node represents a type of bacteria. The size of the nodes represents the relative abundance of the bacteria. The edges indicates the presence of an ecological correlation. The red edge indicates positive correlation. The green edge represents negative correlation. The thickness of the edge indicates the level of correlation. Bacteria of the same cluster are shown in the same color. The bacteria names are shown on the right. (**A**) 0–20 cm soil. (**B**) 20–40 cm soil.

In contrast, the fungal co-occurrence network in the deep soil layer consisted of 66 different genera and 200 edges, and all genera were clustered into ten clusters based on correlation (Figure 12B). Among these clusters, Cluster 6 contained 13 genera, including Cladosporium, Conocybe, and Paecilomyces, which were relatively abundant and correlated with eight, nine, and two different genera, respectively, and were the core nodes in Cluster 6. Cluster 6 was correlated with several other clusters and was at the core position in the co-occurrence network. Cluster 1 contained 11 genera, among which Penicillium and Thermoascus were relatively more abundant. Additionally, Cluster 8, Cluster 9, and Cluster10 all contain only two genera each, with Cluster 8 consisting of Lecanicillium and Didymella, Cluster 9 consisting of Enterocarpus and Talaromyces, and Cluster 10 consisting of Achroceratosphaeria and Tetracladium.

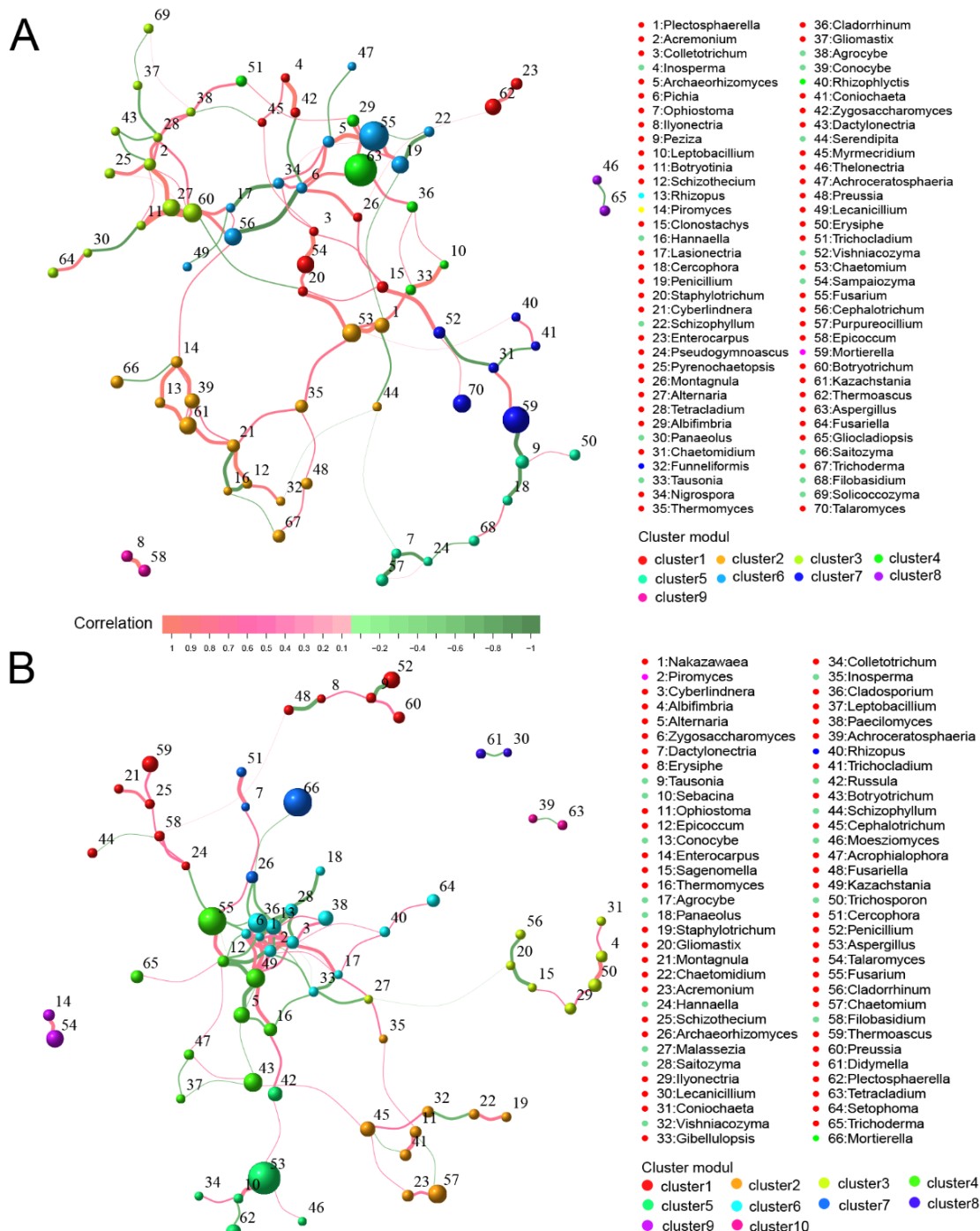

**Figure 12.** Co-occurrence network of fungi. Each node represents a type of bacteria. The size of the nodes represents the relative abundance of the fungi. The edges indicates the presence of an ecological correlation. The red edge indicates positive correlation. The green edge represents negative correlation. The thickness of the edge indicates the level of correlation. Bacteria of the same cluster are shown in the same color. The fungi names are shown on the right. (**A**) 0–20 cm soil. (**B**) 20–40 cm soil.

Overall, these findings suggest that the fungal co-occurrence network in the deep soil layer is more complex and has more inter-correlated clusters compared to the shallow soil layer. Moreover, some of the fungal genera in these clusters may play a crucial role in maintaining the stability and structure of the fungal community in both soil layers.

## 4. Discussion

Soil nitrogen is essential for plant growth, and it mainly exists in the form of $NH_4^+$-N and $NO_3^-$-N. The appropriate nitrogen fertilizer and other components can increase crop production [17]. However, excessive application of nitrogen can cause low fertilizer utilization and environmental pollution, with ammonia volatilization being a critical in nitrogen loss in agricultural fields [18]. The concentration of $NH^{4+}$ in the soil surface rapidly increases after nitrogen fertilizer application, promoting ammonia volatilization, which increases with the amount of nitrogen applied [19]. While most studies have shown that nitrogen additions significantly increase the levels of soils' ammonium and nitrate nitrogen [20], some studies have found that nitrogen does not significantly affect the effective soil TN. In these cases, the fertilizer applied was proportional to $NH_4^+$-N content in the soil surface layer, and high nitrogen application resulted in increased $NH_4^+$-N content in the surface layer, but decreased content in the deeper soil layers [21]; this is similar to the results of the present study. Furthermore, this study concluded that when nitrogen fertilizer was applied too low or too high, the soil TN content was higher in the 0–20 cm soil layer and lower in the 20–40 cm soil layer. This result may be due to high fertilizer application causing enrichment in the 0–20 cm soil layer, while low nitrogen levels result in fertilizer waste due to early crop failure, resulting in less absorption of nitrogen fertilizer.

The impact of nitrogen application on the soil's total phosphorus (TP) and available phosphorus (AP) content is still a matter of debate in the scientific community. While some previous studies have suggested that nitrogen loading has no significant effect on soil TP and AP [22], others have found that N addition significantly decreased soil inorganic phosphor content [23]. On the other hand, experiments on grasslands have indicated that N addition reduced soil TP content, but had no significant effect on the AP content [24].

In the present study, it was found that reducing N application by 20% increased the soil's TP content from 0–40 cm, but N fertilization up to 375 kg/hm$^2$ reduced the soil's AP content in the 0–20 cm soil layer. Additionally, N application increased soil ammonium and nitrate N while causing a significant decrease in soil pH. These findings suggest that the impact of N application on soil TP and AP content may vary depending on factors such as soil type, fertilization rate, and depth of application.

Available potassium in soil is closely related to the in-season potassium supply capacity [25]. The current study found that the available potassium (AK) content in the 0–40 cm soil layer increased initially and then decreased with decreasing N application, while the total potassium (TK) content showed the opposite trend of decreasing then increasing. This may be because potassium in the deep soil replenishes the surface layer after reduced N application, resulting in an integrated effect among soil nutrients to meet the cotton's nutrient demand. Some studies have reported that N fertilization does not affect soil TK and AK [26]. However, in this experiment, both soil TK and AK contents were affected under reduced N application conditions, possibly due to the stimulation of soil potassium after appropriate N fertilization reduction, leading to high AK content in the soil but reduced TK content.

The magnitude of soil microbial diversity and biomass is essential for maintaining farmland microecology. It is generally accepted that excessive nitrogen fertilizer application negatively impacts soil microbial diversity and biomass [27,28]. It has been shown that excessive nitrogen application significantly reduces the species and abundance of microorganisms in soil [29]. However, the results of the present study differed from this. Alpha analysis showed that there was no significant difference in bacterial community diversity between the CK without nitrogen reduction and the groups with substantial nitrogen reduction (N0 and N4), while bacterial diversity in the group with moderate nitrogen reduction (N2) was significantly reduced. This suggests a complex mechanism by which bacterial communities in agricultural soils respond to nitrogen application, on the one hand, nitrogen application not only directly changes microbial distribution, but also indirectly affects soil microbial community structure by changing physicochemical properties such as PH and water content [30,31]. Moreover, farmland microecological environments

vary greatly from soil to soil and have different abilities to maintain microbial community structure homeostasis. On the other hand, the RNE treatment in this study lasted for two years, and short-term changes in nitrogen application may have limited effects on the overall microbial community structure. The β analysis also yielded similar results, with the first and second coordinates in the PCA explaining the majority of the differences, and bacteria more sensitive to environmental disturbances showing significant separation in all samples. However, the overall consistency of fungal community structure was high among treatments, which further proves that short-term reduction of nitrogen fertilizer application can hardly comprehensively affect the microbial community distribution in soil. Both types of research on microbial responses to nitrogen fertilization and agronomic measures to reduce nitrogen require more consideration of long-term benefits.

The present study found that RNE treatments had a significant impact on the abundance of specific microorganisms in the soil, although the overall microbial community structure remained relatively unchanged. Specifically, RNE resulted in an increase in the relative abundance of MND1, RB41 and Sphingomonas. MND1 are a group of ammonia-oxidizing bacteria (AOB) that play an essential role in the soil nitrogen cycling through nitrification and nitrogen fixation in agricultural soils [32]. RB41 is considered a structural and functional cornerstone in plant–soil microbiome and agroecosystems, which can selectively regulate soil organic matter decomposition and carbon cycling in agricultural soil [33,34]. Sphingomonas sp. has been shown to produce plant growth hormone-like substances such as sphingosine sphingan and gellan gum, both of which can improve plant growth and development [35]. The increase in the abundance of these bacteria under N fertilizer reduction may have increased the soil's carbon and nitrogen sequestration potential and improved the farmland's soil status in several ways, including nutrient cycling, soil structure, and biological interactions.

However, not all changes in microbial abundance induced by RNE were beneficial. At a 40% reduction of nitrogen, the relative abundance of Chaetomium and Penicillium decreased, while that of fusarium increased compared to the control. Chaetomium and Penicillium are known as important biocontrol agents that decompose biological residues and protect plants form various pathogens [36]. Penicillium also plays an active role in the soil phosphorus cycle [37], while Fusarium is a major plant pathogenic [38]. These results suggest that excessive N fertilizer reduction may decrease biotic stress tolerance in crops, leading to potential risks for crop production. Therefore, it is crucial to integrate economic and ecological factors and balance RNE with other agricultural inputs in practical production.

The functional analysis of bacteria and fungi further provided evidence for the dual effects of RNE. At the bacterial level, soils with 20% N reduction showed significant enhancement of nitrification and aerobic ammonia oxidation, both of which affect plant nitrogen uptake and utilization Aerobic ammonia oxidation is also highly involved in the carbon and nitrogen cycle in soils, although its significance is still controversial [10,30]. However, at the fungal level, N4 increased the functions associated with phytopathogenic bacteria in the soil, which increased the likelihood of crop disease susceptibility, and decreased the functions of woody saprophytes that primarily degrade plant residues [39]. Although these changes in fungal functions did not reach significance, a significant reduction in nitrogen resulted in a deterioration of the fungal community in the soil.

Core bacterial genera in the microbial communities were identified through multiple correlation analyses. Redundancy analysis revealed that RB41 and MND1 were positively correlated with the microbial structure in shallow soils treated with N2 and N4. Additionally, Ellin6067 is involved in glycolic acid metabolism and defends against harmful pathogens [40], Gemmatimonas in phosphate and hypophosphate metabolism [41], and Sphingomonas in aromatic soil contaminant degradation. These microorganisms were also significantly and positively associated with both RNE treatments. These results suggested that RNE can improve the state of the shallow soil microbial community, with AP playing a major role in this process.

Interestingly, the formation of microbial community structure in both treatments was significantly negatively correlated with nitrate and ammonium nitrogen, suggesting that the interplay between different nutrients may be modulated by affecting microbial community structure variation, which in turn impacts the turnover and transport of different nutrients. In addition to highly abundant microbial genera, the co-occurrence network also revealed many genera with lower abundance but high correlation with various other genera. For example, Nocardia reduces soil cadmium contamination [42]; SWB02 is involved in carbon cycling and is associated with plant-interacting microorganisms [43]; Chaetomium is associated with the metabolism of various substances such as cellulose [44], and Cladosporium is a plant pathogenic fungus [45]. These microorganisms likewise play a crucial role in shaping the community structure.

## 5. Conclusions

Our study shows that short-term nitrogen fertilizer application has altered the physicochemical condition of the soil and partially disturbed the microbial community. Reduction in nitrogen fertilizer increased the relative abundance of MND1 (1.21%), RB41 (1.96%), and Sphingomonas (2.77%), and decreased the relative abundance of Motility East Hidrospermopsis (0.3%). It also reduced the relative abundance of Chaetomium (0.41%) and Penicillium (0.50%). The functional composition of bacteria and fungi in the 0–20 cm soil layer was affected by nitrogen fertilizer reduction, with significant increases in aerobic ammonia oxidation (4.20%) and nitrification (4.10%) and reductions in chemo-heterotrophy (2.70%) and fermentation (4.08%). Phytopathogenic bacteria function increased by 2.74%, while woody saprophytic bacteria function decreased by 11.20% after a 20% nitrogen reduction. Soil $NH^{4+}$-N and PH primarily drove variation in the predominant bacterial community structure. AP specifically drove significant variation in the bacterial community structure under 20% and 40% RNE treatments. Steroidobacter, RB41, Gemmatimonas, Ellin6067, Salinobacterium Haliangium, and Sphingomonas were mainly involved in the formation of bacterial community structure in the 20% and 40% RNE treatments. AP mainly drove variation in fungal community structure. Fusarium, Talaromyces, Penicillium, Saccharomyces, and Chaetomium Sampaiozyma and Chaetomium were mainly involved in the formation of fungal community structures.

**Supplementary Materials:** The following supporting information can be downloaded at: https://www.mdpi.com/article/10.3390/agriculture13040796/s1. Table S1: Cotton production statistics for two years; Table S2: Sequencing quality control statistics.

**Author Contributions:** J.T., C.W. and L.S. performed the experiment and data analysis. J.T. drafted the manuscript. J.W. and J.Z. provided input early in the study. Y.F. and L.M. participated in the writing and revision of the manuscript. W.X. designed the research and revised the manuscript. All authors have read and agreed to the published version of the manuscript.

**Funding:** This study was supported by National Key R&D Program of China (2020YFD1001001).

**Institutional Review Board Statement:** Not applicable.

**Data Availability Statement:** The datasets presented in this study can be found in online repositories. The names of the repository/repositories and accession number(s) can be found in the article/Supplementary Materials.

**Conflicts of Interest:** The authors declare that the research was conducted in the absence of any commercial or financial relationships that could be construed as a potential conflict of interest.

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
