# Peer review of "Moderate Nitrogen Reduction Increases Nitrogen Use Efficiency and Positively Affects Microbial Communities in Agricultural Soils"

_agriculture, doi:10.3390/agriculture13040796_

Round 1

Reviewer 1 Report

The authors report an interesting study on the effects of different soil nitrogen supplementation on the microbiome in cotton crops. The authors applied the latest molecular biology technologies to establish that changes in nitrogen levels have significant effects on soil microbial biodiversity.

The manuscript is very interesting and clear. The topic dealt with could have some applications in the agricultural sector.

However, some improvements should be made to the manuscript.

1) In my opinion, the entire manuscript is very focused on microbial biodiversity and little on the effects on cotton crops. As the authors know, soil microbiomes interact with the root system of plants which benefit from the variety of microbiomes. The authors do not discuss, or discuss little, this aspect. They only report a table in the supplementary materials, where some growth parameters of the cotton plant are indicated. However, the analysis reported on microbiomes is useless if it does not find a specific application on crops.

2) The authors should discuss this recent manuscript in the introduction or discussion: Zangani, E. et al. Nitrogen and Phosphorus Addition to Soil Improves Seed Yield, Foliar Stomatal Conductance, and the Photosynthetic Response of Rapeseed (Brassica napus L.). Agriculture 2021, 11, 483. https://doi.org/10.3390/agriculture11060483

Author Response

Dear Reviewers:

We are honored to have this manuscript reviewed by you. Thank your  comments concerning our manuscript. Those comments are all valuable and very helpful for revising and improving our paper, as well as the important guiding significance to our researches. We have studied comments carefully and have made correction which we hope meet with approval. Revised portion are marked in red in the paper. The main corrections in the paper and the responds to the reviewer’s comments are as flowing:

1. We aimed to investigate the correlation between N fertilization and the global vision of microorganisms in our study. While our findings suggest that some microorganisms such as MND1, RB41, and others may contribute to increased N fertilizer use efficiency, we could not directly demonstrate their impact on cotton yield. This would require the isolation and functional validation of microorganisms, which is beyond the scope of our study. Therefore, we refrained from discussing cotton yield data in our results and instead provided a descriptive study. We hope that you will appreciate our approach and reasoning.

2. We have cited this paper and discussed it as necessary, and appreciate your suggestions for improving our manuscript (page 703)

Reviewer 2 Report

Comments on agriculture-2221396-peer-review-v1 titled “Moderate nitrogen reduction increases nitrogen use efficiency and positively affects microbial communities in agricultural soils  ’’

Abstract:

The results in the abstract section look too general. Pls. clearly describe your results in the abstract and these must be specific acc. to study. Pls. give % increase or decrease or give numerical value for the parameters under study. Pls. clearly state you finding in one sentence in abstract.

Keywords: please don’t repeat words which are included in title

Introduction:

Introduction is too short. Pleas rewrite your introduction and add recent literature regarding your hypothesis. Please add literature regarding nitrogen application, nitrogen use efficiency and its interaction with microbial community.

Materials and methods:

Please add complete crop husbandry in materials and methods section like soil preparation, irrigation and weed management etc

Results

Please describe your variable at first appearance then abbreviate throughout the sentence.

Discussion section is very extensive and should be improved. Please, avoid repetition of results. Discussion should be improved by adding logical reasoning.

The conclusions should answer the hypothesis of your study and should focus on the implication of your findings. Remember that the conclusions must be self-explanatory. This section should highlight the novelty and implication of your study.

Tables: Table 1 please add S.E. to better understand the difference in mean values

Author Response

Dear Reviewers:

We are honored to have this manuscript reviewed by you. Thank your  comments concerning our manuscript. Those comments are all valuable and very helpful for revising and improving our paper, as well as the important guiding significance to our researches. We have studied comments carefully and have made correction which we hope meet with approval. Revised portion are marked in red in the paper. The main corrections in the paper and the responds to the reviewer’s comments are as flowing: 

  1. We substantially improved the entire manuscript to make it more readable and logical.
  2.  We refined the description of the sampling site in the material methodology.
  3.  We added SE to the table1

Reviewer 3 Report

The manuscript must be improved in English. In some points is incomprehensible.

In general, there is no novelty in this manuscript.

In Results section, the description is too complicated in order to understand it.  It is chaotic.

Author Response

Dear Reviewers:

We are honored to have this manuscript reviewed by you. Thank your  comments concerning our manuscript. Those comments are all valuable and very helpful for revising and improving our paper, as well as the important guiding significance to our researches. We have studied comments carefully and have made correction which we hope meet with approval. Revised portion are marked in red in the paper. The main corrections in the paper and the responds to the reviewer’s comments are as flowing: 

1. We have comprehensively improved the English of the manuscript, enhancing the readability of the article.

2. We have reworked the content of the Results section to clarify the logical relationships.

Round 2

Reviewer 1 Report

The authors have significantly improved the manuscript. In my opinion, the manuscript can be published.

Author Response

Thank you for your appreciation. Once again, we appreciate your guidance and thank you for your time and effort in reviewing our manuscript.

Reviewer 2 Report

Pls. recheck table as in some mean values S.E. is zero. There may be some non-zero value 

Author Response

Dear Reviewers,

Thank you for your valuable feedback on our manuscript. We have carefully considered your comments and have made the necessary revisions to improve the quality of our research.

In response to your previous concerns about the SE values being too small and recorded as 0.0000, we have now updated our results. The SE values of the Simpson are statistically significant and have been retained with five decimal places. We have highlighted these values in table 1 to improve their visibility and clarity for readers.

Once again, we appreciate your guidance and thank you for your time and effort in reviewing our manuscript.

Best regards

Reviewer 3 Report

In general, the manuscript and english language hane been signifiacantly improved. 

Author Response

(The authors gave the same response as above.)
